# Probing LLMs for hate speech detection: strengths and vulnerabilities

**Sarthak Roy, Ashish Harshavardhan, Animesh Mukherjee** and **Punyajoy Saha**

Indian Institute of Technology, Kharagpur

{sarthak.cse22@kgpian,animeshm@cse}.iitkgp.ac.in,

{ashishaug29,punyajoys}@iitkgp.ac.in

## Abstract

Recently efforts have been made by social media platforms as well as researchers to detect hateful or toxic language using large language models. However, none of these works aim to use explanation, additional context and victim community information in the detection process. We utilise different prompt variation, input information and evaluate large language models in zero shot setting (without adding any in-context examples). We select three large language models (GPT-3.5, text-davinci and Flan-T5) and three datasets - HateXplain, implicit hate and ToxicSpans. We find that on average including the target information in the pipeline improves the model performance substantially ($\sim 20-30\%$) over the baseline across the datasets. There is also a considerable effect of adding the rationales/explanations into the pipeline ($\sim 10-20\%$) over the baseline across the datasets. In addition, we further provide a typology of the error cases where these large language models fail to (i) classify and (ii) explain the reason for the decisions they take. Such vulnerable points automatically constitute 'jailbreak' prompts for these models and industry scale safeguard techniques need to be developed to make the models robust against such prompts.

## 1 Introduction

Abusive language has become a perpetual problem in today's online social media. An ever-increasing number of individuals are falling prey to online harassment, abuse and cyberbullying as established in a recent study by Pew Research (Vogels, 2021). In the online setting, such abusive behaviours can lead to traumatization of the victims (Vedeler et al., 2019), affecting them psychologically. Furthermore, widespread usage of such content may lead to increased bias against the target community, making violence normative (Luft, 2019). Many gruesome incidents like the mass shooting at Pitts-

burgh synagogue[1], the Charlottesville car attack[2], etc. have all been caused by perpetrators consuming/producing such abusive content.

In response to this, social media platforms have implemented moderation policies to reduce the spread of hate/abusive content. One important step in content moderation is filtering of abusive content. A common way to do this is to train language models on human annotated contents for classification. However there are challenges in this approach in the forms of heavy resources required in terms of labour and expertise to annotate these hateful contents. This exercise also exposes the annotators to a wide array of hateful contents that is almost always psychologically very taxing. Therefore, recently, many works have tried to understand if Large language models (LLMs) can be used for detecting such abusive language[3] (Huang et al., 2023; Ziems et al., 2023), but none of these study the role of additional context as input (output) to (from) such LLMs.

We, for the first time, introduce several prompt variations and input instructions to probe two of the LLMs (GPT 3.5 and text-davinci) across three datasets - HateXplain (Mathew et al., 2021), implicit hate (ElSherief et al., 2021) and Toxic-Spans (Pavlopoulos et al., 2022). Note that all these three datasets contain ground truth explanations in the form of either rationales (Mathew et al., 2021; Pavlopoulos et al., 2022) or implied statements (ElSherief et al., 2021) that tells why an annotator took a particular labelling decision. In addition, two of the datasets also contain the information about the target/victim community against whom the hate speech was hurled. In particular, we design prompts that contain (a) only the hate post as input and a query for the output label (vanilla)

---

[1] https://en.wikipedia.org/wiki/Pittsburgh_synagogue_shooting

[2] https://en.wikipedia.org/wiki/Charlottesville_car_attack

[3] https://tinyurl.com/detoxigen

(b) post as well as the definition of hate speech as input and a query for the output label (c) post (-/+ definition) and the target community information as input and a query for the output label, (d) post (-/+ definition) and the explanation as input and a query for the output label, (e) post (-/+ definition) as input and a query for the output label and the target community, and (f) post (-/+ definition) as input and a query for the output label and the explanation. We record the performance of all these approaches and also identify the most confusing cases. In order to facilitate future research, we further provide a typology of the error cases where the LLMs fail to classify and usually provide poor explanations for the classification decisions taken. This typology would naturally constitute the 'jailbreak' prompts[4] to which such LLMs are vulnerable thus pointing to the exact directions in which industry scale safeguards need to be built.

We make the following observations.

- In terms of vanilla prompts (that is case (a)), we find that `flan-T5-large` performs the best among the three models; we also observe that `text-davinci-003` is better than `gpt-3.5-turbo-0301` although the latter is a more recent version.

- Our proposed strategies of prompts individually benefit the LLMs in most cases. The prompt with target community in the pipeline gives the best performance with $\sim 20-30\%$ improvements over the vanilla setup. None of these LLMs are able to benefit themselves if multiple prompt strategies are combined.

- While doing a detailed error analysis, we find that the misclassification of non-hate/non-toxic class is the most common error for implicit hate and ToxicSpans datasets while for the Hatexplain dataset the majority of misclassifications are from the normal to the offensive class. There are also a large number of cases where the model confuses between the hate speech and the offensive class.

- From the typology induced from the error cases, we find many interesting patterns. These LLMs make errors due to the presence of sensitive or controversial terms in otherwise non-hateful posts. Presence of negation words and words expressing support for a community are misclassified as hateful. Ideological

posts and posts containing opinions or fact check information about news articles are often misclassified as toxic/hateful. On the other hand, many offensive/hateful posts are marked normal by the model either due to a vocabulary gap or presence of unknown or polysemous words. Similarly, these models miss to classify implicitly toxic posts and mark them as non-toxic.

We make our codes and resources used for this research publicly available for reproducibility purposes[5].

## 2 Related works

*Large language models*: Based on the training setup, the model architecture and the use cases LLMs can be broadly classified into encoder-only, encoder-decoder based and decoder-only types. In recent years decoder-only LLMs[6] have seen a huge surge with industry scale releases like chatGPT, gpt-4, BARD, LLaMa, etc. Decoder-only LLMs have been used for benchmarks like GLUE (Zhong et al., 2023) where there is no downstream application involved. In the classification setting LLMs have been extensively used for sentiment analysis (Zhang et al., 2023b). They have also been heavily used in NLI and QA tasks on multiple datasets (Chowdhery et al., 2022). In the generation setting, LLMs have found applications in summarization (Zhang et al., 2023a), machine translation (Chowdhery et al., 2022) and open-ended generation (Brown et al., 2020).

*LLMs for hate speech detection*: In (Zhu et al., 2023), the authors use chatGPT to relabel various datasets one of which is on hate speech detection and found that the agreement with human annotations is still quite poor. The authors in (Li et al., 2023) use chatGPT to classify a comment as harmful (i.e., hateful, offensive, or toxic – HOT) and found that the model is better in identifying non-HOT comments than HOT comments. Finally in (Huang et al., 2023) the authors attempted to classify implicit hate speech using chatGPT. However, their prompt was framed as a 'yes/no' question (rather than based on the exact classes i.e., implicit hate, explicit hate, non-hate as in the original study (ElSherief et al., 2021)) which makes the problem lose its original fervour.

---

[4] `https://docs.kanaries.net/articles/chatgpt-jailbreak-prompt`

[5] `https://shorturl.at/nqTX2`
[6] `https://github.com/Hannibal046/Awesome-LLM`

## 3 Datasets and metrics

For all these datasets, we utilise (create) a test dataset for our experiments.

**Implicit hate dataset**: The implicit hate (ElSherief et al., 2021) corpus is a specialized collection of data aimed at detecting hate speech. It provides detailed labels (implicit_hate, explicit_hate or non_hate) for each message, including information about the implied meaning behind the content. The dataset comprises 22,056 tweets sourced from major extremist groups in the United States.

We test on a subset of 2147 samples (108 entries labelled explicit_hate, 710 entries labelled implicit_hate, and 1329 entries labelled not_hate) which are sampled from the entire dataset in a stratified fashion. Note that we do not have explanations and targets for all the posts. The implied statements and targets were available only for the samples with label implicit_hate. Hence, when we experiment with explanation as input (see section 5), we pass the input post as the implied statement. This is for the explicit_hate and not_hate data points. Both these categories do not required any additional implied statement as there is nothing implied in them. In case of targets as inputs (see section 5), we remove the explicit_hate datapoints since they are targetting some victim community but the targets are not present in the annotated dataset. The targets for not_hate are set as 'none'.

**HateXplain dataset**: HateXplain (Mathew et al., 2021) is a benchmark dataset specifically designed to address bias and explainability in the domain of hate speech. It provides comprehensive annotations for each post, encompassing three key perspectives: classification (hate speech, offensive, or normal), the targeted community, and the rationales - which denote the specific sections of a post that influenced the labelling decision (hate, offensive, or normal). We test on the already released test dataset containing 1924 samples (594 entries labelled as hate speech, 782 entries labelled as normal, and 548 entries labelled as offensive). Note that we do not have rationales for the normal posts. In the explanations as input experiments (see section 5), the complete post (tokenized) is taken as their rationale for the normal posts.

**ToxicSpans dataset**: The ToxicSpans (Pavlopoulos et al., 2022) dataset is a subset (containing 11,006 samples labelled toxic) of the Civil Comments dataset (1.2M Posts). The dataset also contains the toxic spans, i.e., the region of the texts found toxic. Not all the posts had toxic spans annotated. We create a test set of 2000 samples by picking 1000 samples labelled toxic from this dataset (where the toxic spans were available), and 1000 samples labelled non-toxic from the Civil Comments dataset (Borkan et al., 2019). Note that we have the spans/rationales marked only for the toxic data points. For the non-toxic posts For explanations as input experiments (see section 5), the complete post (tokenized) is taken as the rationale.

**Metrics**: For primary evaluation, we rely on classification performance. We use precision, recall, accuracy and macro F1-score to measure classification performance, which are all standard metrics.

The data points which have rationales are additionally evaluated using other generation metrics. For the natural language explanations (implicit hate dataset) we use `BERTScore` averaged over all data points. `BERTScore` (Zhang et al.) computes a similarity score for each token in the candidate sentence with each token in the reference sentence. However, instead of exact matches, they compute token similarity using contextual embeddings. For extractive explanation (HateXplain/ToxicSpans dataset), we use average `sentence-BLEU` (Papineni et al., 2002) score which is the standard among the generation metrics.

## 4 Models

For our experiments we utilise three LLMs. Two of these are from the proprietary GPT-3.5 model series[7] while the third is an open source one from the T5 series.

GPT-3.5 models are better than their predecessors GPT 3 (Brown et al., 2020). Both of these models are highly advanced language models capable of generating human-like text based on the provided prompts but they differ in some key ways. As per their documentation[8], GPT-3 was optimized on code completion tasks to create `code-davinci-002`. This was further improved using instruction finetuning (Ouyang et al., 2022) to create `text-davinci-002`. This was later upgraded to `text-davinci-003` which was trained on a larger dataset (ope) making it better at higher quality text generation, following instructions and generating longer context. `gpt-3.5-turbo-0301` is an improvement over `text-davinci-003`. We

---

[7]This was made available by the Microsoft Accelerating AI Academic Research grant.

[8]https://platform.openai.com/docs/model-index-for-researchers

choose these two models in our study namely the `gpt-3.5-turbo` and the `text-davinci-003`.

The third model we use is the open source `flan-T5-large` which is an instruction finetuned variant of the popular T5 model (Chung et al., 2022). As per their documentation the model was instruction finetuned with an emphahsis on scaling the number of tasks, scaling the model size and introducing chain of thought data in the finetuning pipeline. The authors have claimed that this particular sort of finetuning has benefited the T5 models greatly by outperforming models of higher size like GPT-3.

## 5 Prompts

In this section, we list the prompt variations used in this work. A concise summary of the different variants is noted in the Appendix A (Table 7) and the details for each are discussed in the subsections below.

| Dataset | Label list |
|---------|-----------|
| HateXplain | normal, offensive or hate speech |
| Implicit hate | explicit_hate, implicit_hate, or not_hate |
| ToxicSpans | toxic or non_toxic |

Table 1: The list of labels for each dataset.

**Vanilla prompts**: In this category, we use a prompt template where we ask the model to classify the given `post` into a label out of a `list_of_labels`. In addition, we also provide a few `example_outputs` (one class per line) for helping the models generate proper answer. The `list_of_labels` for each datasets are noted in the Table 1.

**(+) definitions**: In the vanilla prompts we assumed that the LLMs are to an extent aware of the labels for classification. Here, we provide the definitions as an additional context to the LLMs. This can help the LLMs understand the classification tasks better. These definitions are added as a list where each label's definition is separated by a new line. We note this prompt template as **Vanilla + Defn** in Appendix A, Table 7. Individual definitions of the tasks are added in Appendix B.

**(+) explanations** Recently, there has been a huge interest in developing explainable deep learning models where the prediction decision is supported by an explanation (Bhatt et al., 2020). We test two hypotheses – (a) whether providing explanations to LLMs as inputs (corresponding to the templates **Vanilla + Exp (input)** and **Vanilla + Defn +**

**Exp (input)**) improve its labelling decisions and (b) whether asking LLMs for an explanation about its labelling decision forces it to predict better labels and as well generate relevant explanations (corresponding to the templates **Vanilla + Exp (output)** and **Vanilla + Defn + Exp (output)**). For the HateXplain and the ToxicSpans dataset the ground truth explanations are in the form of rationales (i.e., a part (parts) of the post that the annotator marked as the reason for his/her labelling decision). For the implicit hate speech dataset the ground truth explanations are in the form of implied statements. In the templates **Vanilla + Exp (output)** and **Vanilla + Defn + Exp (output)** we use two variables `explanation_type` and `explanation_format` (see Appendix A, Table 7). For each dataset, we note the values in these lists below.

- *HateXplain*: Here, `explanation_type` is "extract the words from the post that you found as hate speech or offensive" and `explanation_format` is "the list of extracted words, separated by ". Enclose the list with < < < > > >"
- *ToxicSpans*: Here, `explanation_type` is "extract the words from the post that you found as toxic" and `explanation_format` is "the list of extracted words, separated by ". Enclose the list with < < < > > >"
- *Implicit hate*: Here, `explanation_type` is "with an explanation in 15 words" and `explanation_format` is "the explanation enclosed in < < < > > >"

In the templates **Vanilla + Exp (input)** and **Vanilla + Defn + Exp (input)** we use a single variable `explanation` (see Appendix A, Table 7). For each dataset, we note the value in this list below.

- *HateXplain*: `explanation` is "the rationales {rationales} as an explanation".
- *ToxicSpans*: `explanation` is "the span {span} as an explanation".
- *Implicit hate*: `explanation` is "the implied statement {implied statement} as an explanation".

**(+) targets** A very important information in any hate speech detection pipeline is the victim community the abusive/hate speech targets. Here again we test two hypotheses – (a) whether providing target to LLMs as inputs (corresponding to the templates **Vanilla + Tar (input)** and **Vanilla + Defn + Tar (input)**) improve its labelling decisions and

| Datasets | F1-Score | Precision | Recall |
|---|---|---|---|
| HateXplain | 0.698 | 0.687 | NR |
| Implicit hate (full) | 0.54 | 0.64 | 0.47 |
| ToxicSpans | 0.31 | 0.79 | 0.20 |

Table 2: BERT-Hatexplain inference on three datasets used to prompt the LLMs. NR: not reportedin the original paper (Mathew et al., 2020).

(b) whether asking LLMs for the target information forces it to predict better labels and as well generate correct targets (corresponding to the templates **Vanilla + Tar (output)** and **Vanilla + Defn + Tar (output)**). In the templates **Vanilla + Tar (output)** and **Vanilla + Defn + Tar (output)** we use two variables `target_type` and `target_format` (see Appendix A, Table 7). For all the datasets, we replace the variable `target_type` with "also mention which group of people does it target" and the variable `target_format` with "list targeted groups enclosed in < < < > > >". In the templates **Vanilla + Tar (input)** and **Vanilla + Defn + Tar (input)** we replace the variable `targets` (see Appendix A, Table 7) with the ground truth target community information which is only applicable for the HateXplain and the implicit hate speech datasets.

## 6 Results

In this section, we note the results of the different prompt variations on the models for the three datasets. As a baseline, we consider BERT-HateXplain (Mathew et al., 2020) and run the model on the implicit hate and ToxicSpan datasets. We have taken the results for the HateXplain dataset from the original paper. Table 2 shows the results of this baseline.

Our key results are noted in Tables 3, 4 and 5 for the HateXplain, implicit hate and ToxicSpans dataset respectively.

**Vanilla prompts**: In terms of vanilla prompts, `flan-T5-large` performs better than `gpt-3.5-turbo-0301` and `text-davinci-003` for the HateXplain and implicit hate (0.59 and 0.63 F1-scores respectively) datasets. Nevertheless, it performs at par with the other two models for the ToxicSpans dataset. Among the larger models, we find that `text-davinci-003` is better than the `gpt-3.5-turbo-0301` in all the datasets in terms of macro F1-score, reporting 15.38%, 12.50% and 4.41% higher values for HateXplain, implicit hate, and ToxicSpans respectively. Overall the performance for the HateXplain (0.39 in `gpt-3.5-turbo-0301` and

0.45 in `text-davinci-003`) and the implicit hate (0.32 in `gpt-3.5-turbo-0301` and 0.36 in `text-davinci-003`) datasets are much less than the ToxicSpans (0.68 in `gpt-3.5-turbo-0301` and 0.71 in `text-davinci-003`) dataset.

**(+) definitions**: Adding definitions to the prompts does not always help in improving the performance. In terms of F1-score, for HateXplain, we see an improvement of 25.64% for `gpt-3.5-turbo-0301` while the performance worsens by 17.78% for `text-davinci-003` and 20.34% for `flan-T5-large`. The situation is reverse for the ToxicSpans dataset where we see an improvement of 1.40% for `text-davinci-003` and 13.75% for `flan-T5-large` while it worsens by 7.35% for `gpt-3.5-turbo-0301`. For ToxicSpans, adding definitions to the input prompt gives the best performance across all the prompting strategies. For implicit hate, there is an improvement of 12.50%, 16.67% and 4.76% for `gpt-3.5-turbo-0301`, `text-davinci-003` and `flan-T5-large` respectively.

**(+) explanations**: As discussed earlier, we exploit the power of explanations/rationales in two ways. For the case where we ask the model to generate explanations along with the label in the output, we see a similar trend like adding definitions. In terms of F1-score, for HateXplain, we see an improvement of 23.07% for `gpt-3.5-turbo-0301` while the performance worsens by 8.89% for `text-davinci-003` and 10.17% for `flan-T5-large`. The situation is similar for the ToxicSpans dataset we see comparable performance for `gpt-3.5-turbo-0301` while it worsens by 9.86% for `text-davinci-003` but it improves by 18.84 % for `flan-T5-large`. For implicit hate, there is improvement of 6.25% and 22.22% for `gpt-3.5-turbo-0301` and `text-davinci-003` respectively but it worsens by 11.11% for `flan-T5-large`. We further compare the generated explanations with the already available ground truth explanations using `BERTScore` for the implicit hate dataset and `sentence-BLEU` for the other two datasets. We observe that the `gpt-3.5-turbo-0301` model generates better explanations (averaged over all data points) than the `text-davinci-003` model for the HateXplain and the implicit hate datasets. For the ToxicSpans dataset the results are reversed.

When we add the respective explanations/rationales in the input as prompts, in

terms of F1-score, for HateXplain the performance is 12.82% better for `gpt-3.5-turbo-0301` and 15.55% better for `text-davinci-003` and comparable for `flan-T5-large`. For implicit hate the results are 3.03% better for `gpt-3.5-turbo-0301` and 8.33% better for `text-davinci-003` but worsens by 14.29% for `flan-T5-large`. The F1-scores for ToxicSpans dataset are 8.11% better for `gpt-3.5-turbo-0301` and 20.29% for `flan-T5-large` but significantly worse ($\sim$9.86%) for `text-davinci-003`.

**(+) targets**: We exploit the power of targets again in two ways. For the case, where we ask the model to generate target along with the label in the output, we again see a similar trend like adding definitions. In terms of F1-score, for HateXplain, we see an improvement of 28.20% for `gpt-3.5-turbo-0301` while the performance worsens by 15.56% for `text-davinci-003` and 11.86% for `flan-T5-large`. The situation is similar for the ToxicSpans dataset we see comparable performance for `gpt-3.5-turbo-0301` while it worsens by 11.27% for `text-davinci-003` and 18.84% for `flan-T5-large`. For implicit hate, there is improvement of 9.38% and 30.56% for `gpt-3.5-turbo-0301` and `text-davinci-003` respectively and comparable results for `flan-T5-large`.

For the case, where we use the target as inputs in the prompt we see improvement for both the datasets where the target is present in the ground truth except for `flan-T5-large`. In terms of F1-score, for HateXplain, we see an improvement of 33.33% for `gpt-3.5-turbo-0301` and 26.67% for `text-davinci-003` and a drop in performance by 6.78% for `flan-T5-large`. For implicit dataset, the ground truth targets are present only for the implicit hate data points. Thus while comparing with the **Vanilla** setup, we only consider the implicit hate data points to have a fair comparison. The revised F1 scores for the **Vanilla** setup become 0.52 for `gpt-3.5-turbo-0301` and 0.68 for `text-davinci-003`. Comparison with this revised values show an improvement of 17.30% for `gpt-3.5-turbo-0301` while for `text-davinci-003` the results remain roughly unchanged. In case of `flan-T5-large` the performance drops by 9.52%. For ToxicSpans we do not have the target annotated and thus we skip this experiment for this dataset. Overall target as inputs outperforms most of the other prompt strategies

across both these models and the two datasets. This leads us to believe that future annotation exercises for training hate speech detection models should almost always benefit if the target information is also annotated.

**Combinations**: Next, we evaluate the cases when we utilise definitions as well as another additional prompt strategy (target/explanation at input/output). The performance of adding explanations either at input/output along with definitions does not help the models much with the performance remaining comparable, e.g., for `gpt-3.5-turbo-0301` on HateXplain (+) explanation (input) with definition vs with (+) explanation (input) or worsens e.g., for `gpt-3.5-turbo-0301` on HateXplain (+) explanation (output) with definition vs with (+) explanation (output). The quality of the explanations generated (in terms of average BERTScore or sentence-BLEU) are almost always worse in presence of the definitions. worse Similar situation arises when we add definitions with target (input/output) where the performance either remains comparable or worsens. Only in the case of the ToxicSpans dataset, we see that adding both explanation (input) and definition for the `gpt-3.5-turbo-0301` and `flan-T5-large` model and gives the best performance.

# 7 Error analysis and attack points

In this section we shall first discuss the cases where the models encounter a large number of misclassifications and then outline a technique to induce a typology of the different attack points to which the models are vulnerable.

**Significance test**: We compare the results of the vanilla model for each dataset with the best performing prompt plus model combination using the Mann-Whitney U test. We find that the results are significant. We next present the $p$-values for each $-$ (a) `gpt-3.5-turbo-0301` + HateXplain: $p = 0.0042$ (vanilla vs target-at-input), (b) `gpt-3.5-turbo-0301` + implicit hate: $p = 0.0$ (vanilla vs target-at-input), (c) `gpt-3.5-turbo-0301` + ToxicSpans: $p = 0.00088$ (vanilla vs explanation-at-input), (d) `text-davinci-003` + HateXplain: $p = 9.946e - 1$ (vanilla vs target-at-input), (e) `text-davinci-003` + implicit hate: $p = 4.441e - 16$ (vanilla vs target-at-input), (f) `text-davinci-003` + ToxicSpans: $p = 0.01524$ (vanilla vs definition-at-input), (g) `flan-T5-large` + HateXplain: $p = 0.0$ (vanilla vs

| Model | Strategies | | | | | Metrics | | | | |
|---|---|---|---|---|---|---|---|---|---|---|
| | Ex (o) | Ta (o) | D | Ex (i) | Ta (i) | Acc | Pre | Rec | F1 | BS/BL |
| GPT-3.5 | | | | | | 0.45 | 0.54 | 0.46 | 0.39 | |
| | ✓ | | | | | 0.50 | 0.56 | 0.52 | 0.48 | **0.15** |
| | | | ✓ | | | 0.50 | **0.63** | 0.53 | 0.49 | |
| | ✓ | | ✓ | | | 0.49 | 0.60 | 0.52 | 0.49 | 0.13 |
| | | | | ✓ | | 0.48 | 0.55 | 0.49 | 0.44 | |
| | | | | | ✓ | **0.53** | 0.60 | **0.56** | **0.52** | |
| | | | | ✓ | ✓ | 0.52 | 0.60 | 0.55 | 0.50 | |
| | | | ✓ | | ✓ | 0.52 | **0.63** | 0.55 | 0.51 | |
| | | ✓ | | | | 0.52 | 0.61 | 0.55 | 0.50 | |
| | | ✓ | ✓ | | | 0.49 | **0.63** | 0.53 | 0.47 | |
| Davinci | | | | | | 0.47 | 0.55 | 0.47 | 0.45 | |
| | ✓ | | | | | 0.46 | 0.53 | 0.44 | 0.41 | **0.05** |
| | | | ✓ | | | 0.44 | 0.58 | 0.44 | 0.37 | |
| | ✓ | | ✓ | | | 0.41 | 0.57 | 0.41 | 0.34 | 7e-4 |
| | | | | ✓ | | 0.46 | 0.56 | 0.45 | 0.41 | |
| | | | | | ✓ | 0.53 | 0.57 | 0.51 | 0.52 | |
| | | | | ✓ | ✓ | **0.57** | 0.62 | **0.57** | **0.57** | |
| | | | ✓ | | ✓ | 0.54 | **0.64** | 0.53 | 0.53 | |
| | | ✓ | | | | 0.44 | 0.55 | 0.43 | 0.38 | |
| | | ✓ | ✓ | | | 0.41 | 0.49 | 0.42 | 0.33 | |
| Flan-T5 | | | | | | 0.60 | **0.70** | **0.65** | 0.59 | |
| | ✓ | | | | | 0.54 | 0.63 | 0.60 | 0.53 | **0.12** |
| | | | ✓ | | | 0.51 | 0.67 | 0.58 | 0.47 | |
| | ✓ | | ✓ | | | 0.57 | 0.57 | 0.57 | 0.57 | 0.02 |
| | | | | ✓ | | 0.54 | 0.69 | 0.60 | 0.51 | |
| | | | | | ✓ | 0.60 | 0.62 | 0.62 | 0.60 | |
| | | | | ✓ | ✓ | 0.56 | 0.64 | 0.61 | 0.55 | |
| | | | ✓ | | ✓ | 0.62 | 0.67 | **0.65** | **0.62** | |
| | | ✓ | | | | **0.64** | **0.70** | 0.57 | 0.52 | |
| | | ✓ | ✓ | | | 0.62 | 0.61 | 0.61 | 0.61 | |

Table 3: Results for the HateXplain dataset with different prompt variations. Ex: explanation, Ta: target, D: definition, i: input, o: output. Acc: accuracy, Pre: precision, Rec: recall, BS: BERTScore, BL: sentence-BLEU. The best results are highlighted.

| Model | Strategies | | | | | Metrics | | | | |
|---|---|---|---|---|---|---|---|---|---|---|
| | Ex (o) | Ta (o) | D | Ex (i) | Ta (i) | Acc | Pre | Rec | F1 | BS/BL |
| GPT-3.5 | | | | | | 0.35 | 0.45 | 0.51 | 0.32 | |
| | ✓ | | | | | 0.36 | 0.46 | 0.52 | 0.34 | **0.85** |
| | | | ✓ | | | 0.41 | 0.49 | 0.47 | 0.36 | |
| | ✓ | | ✓ | | | 0.41 | 0.49 | 0.49 | 0.36 | **0.85** |
| | | | | ✓ | | 0.37 | 0.44 | 0.41 | 0.33 | |
| | | | | ✓ | ✓ | 0.45 | 0.45 | 0.40 | 0.34 | |
| | | | | | ✓ | **0.61** | 0.68 | **0.68** | **0.61** | |
| | | | ✓ | | ✓ | 0.56 | **0.69** | 0.65 | 0.55 | |
| | | ✓ | | | | 0.38 | 0.47 | 0.49 | 0.35 | |
| | | ✓ | ✓ | | | 0.38 | 0.48 | 0.45 | 0.33 | |
| Davinci | | | | | | 0.52 | 0.44 | 0.54 | 0.36 | |
| | ✓ | | | | | 0.50 | 0.47 | 0.54 | 0.44 | **0.79** |
| | | | ✓ | | | 0.51 | 0.44 | 0.52 | 0.42 | |
| | ✓ | | ✓ | | | 0.36 | 0.42 | 0.50 | 0.32 | 0.78 |
| | | | | ✓ | | 0.45 | 0.40 | 0.45 | 0.39 | |
| | | | | ✓ | ✓ | 0.44 | 0.46 | 0.46 | 0.42 | |
| | | | | | ✓ | **0.73** | **0.70** | **0.68** | **0.68** | |
| | | | ✓ | | ✓ | **0.73** | **0.70** | 0.66 | 0.67 | |
| | | ✓ | | | | 0.61 | 0.48 | 0.56 | 0.47 | |
| | | ✓ | ✓ | | | 0.45 | 0.45 | 0.53 | 0.39 | |
| Flan-T5 | | | | | | 0.67 | 0.64 | 0.62 | 0.63 | |
| | ✓ | | | | | 0.66 | 0.66 | 0.58 | 0.56 | **0.86** |
| | | | ✓ | | | **0.69** | **0.67** | **0.66** | **0.66** | |
| | ✓ | | ✓ | | | 0.67 | 0.66 | **0.66** | **0.66** | 0.83 |
| | | | | ✓ | | 0.64 | 0.62 | 0.54 | 0.50 | |
| | | | | ✓ | ✓ | 0.64 | 0.61 | 0.56 | 0.54 | |
| | | | | | ✓ | 0.64 | 0.58 | 0.57 | 0.57 | |
| | | | ✓ | | ✓ | 0.62 | 0.56 | 0.55 | 0.54 | |
| | | ✓ | | | | 0.63 | 0.64 | 0.65 | 0.63 | |
| | | ✓ | ✓ | | | 0.67 | 0.64 | 0.61 | 0.61 | |

Table 4: Results for the implicit hate dataset with different prompt variations. Ex: explanation, Ta: target, D: definition, i: input, o: output. Acc: accuracy, Pre: precision, Rec: recall, BS: BERTScore, BL: sentence-BLEU. The best results are highlighted.

definition+target-at-input), (h) flan-T5-large + implicit hate: $p = 0.000358$ (vanilla vs definition-at-input), (i) flan-T5-large + ToxicSpans: $p = 0.0$ (vanilla vs explanation -at-input).

**Cases of misclassification**: For the implicit hate dataset we observe that in case of the gpt-3.5-turbo-0301 model, for all the different prompt variants the largest number of misclassifications is from the non-hate to the implicit hate class. For the text-davinci-003 model the major observations are as follows. In the vanilla setup (with or without definition), the implicit hate class is most often confused with the explicit hate class. However, if the prompt has an explanation component (either at input or at output), once again, there is a large number of misclassifications from the non-hate to the implicit hate class. For flan-T5-large we observe that the model fails to classify the implicit hate speech class. The implicit hate is classified as non-hate or explicit hate following the trend of the other two models.

For the HateXplain dataset we observe that in case of the gpt-3.5-turbo-0301 model, for all the different prompt variants the largest number of misclassifications is from the normal to the offensive class. Another confusion that the model often faces is between the hate and the offensive class; if the prompts do not contain the definition (of hate/offensive speech), then offensive speech is largely mislabelled as hate speech while the results are exactly reversed if the prompts contain the definition. For the text-davinci-003 model once again we observe for all the different prompt variants the largest number of misclassifications is from the normal to the offensive class. Further, irrespective of whether the prompts contain the definition or not, hate speech is heavily mislabelled as offensive speech. For flan-T5-large both offensive and hatespeech classes are missclassified as normal speech for all prompt variations.

For the ToxicSpans dataset in case of the gpt-3.5-turbo-0301 model, the non-toxic data points are heavily misclassified as toxic for all prompt variants. Curiously, adding definition to the prompts increases this misclassification. These observations remain similar for the

| Model | Strategies | | | | | Metrics | | | | |
|---|---|---|---|---|---|---|---|---|---|---|
| | Ex (o) | Ta (o) | D | Ex (i) | Ta (i) | Acc | Pre | Rec | F1 | BS/BL |
| GPT-3.5 | | | | | | 0.69 | 0.73 | 0.69 | 0.68 | |
| | ✓ | | | | | 0.70 | 0.75 | 0.70 | 0.68 | **0.26** |
| | | | ✓ | | | 0.67 | 0.75 | 0.67 | 0.63 | |
| | ✓ | | ✓ | | | 0.66 | 0.76 | 0.66 | 0.72 | 0.20 |
| | | | | ✓ | | **0.75** | 0.78 | **0.75** | **0.74** | |
| | | | | ✓ | ✓ | 0.71 | **0.79** | 0.71 | 0.69 | |
| | | ✓ | | | ✓ | - | - | - | - | |
| | | ✓ | | | | 0.70 | 0.74 | 0.70 | 0.69 | |
| | | ✓ | ✓ | | | 0.70 | 0.75 | 0.70 | 0.68 | |
| Davinci | | | | | | 0.71 | 0.71 | 0.71 | 0.71 | |
| | ✓ | | | | | 0.66 | 0.69 | 0.66 | 0.64 | **0.45** |
| | | | ✓ | | | **0.72** | **0.72** | **0.72** | **0.72** | |
| | ✓ | | ✓ | | | 0.71 | 0.71 | 0.71 | 0.71 | 0.35 |
| | | | | ✓ | | 0.65 | 0.66 | 0.65 | 0.64 | |
| | | | | ✓ | ✓ | 0.68 | 0.69 | 0.68 | 0.68 | |
| | | ✓ | | | ✓ | - | - | - | - | |
| | | ✓ | | | | 0.66 | 0.69 | 0.66 | 0.64 | |
| | | ✓ | ✓ | | | 0.69 | 0.71 | 0.69 | 0.68 | |
| Flan-T5 | | | | | | 0.70 | 0.76 | 0.70 | 0.69 | |
| | ✓ | | | | | 0.78 | **0.84** | 0.78 | 0.82 | 0.35 |
| | | | ✓ | | | 0.80 | 0.81 | 0.80 | 0.80 | |
| | ✓ | | ✓ | | | 0.82 | **0.84** | 0.82 | 0.82 | **0.43** |
| | | | | ✓ | | **0.83** | **0.84** | **0.83** | **0.83** | |
| | | | | ✓ | ✓ | **0.83** | 0.83 | **0.83** | **0.83** | |
| | | ✓ | | | ✓ | - | - | - | - | |
| | | ✓ | | | | 0.62 | 0.76 | 0.62 | 0.56 | |
| | | ✓ | ✓ | | | 0.79 | 0.80 | 0.79 | 0.78 | |

Table 5: Results for the ToxicSpans dataset with different prompt variations. Ex: explanation, Ta: target, D: definition, i: input, o: output. Acc: accuracy, Pre: precision, Rec: recall, BS: BERTScore, BL: sentence-BLEU. The best results are highlighted.

text-davinci-003 model. For flan-T5-large non-toxic data points are misclassified as toxic for all the prompt variations except for when target is asked to be generated in which case toxic data-points are misclassified as non-toxic.

In order to better elicit the above observations, we present the confusion matrices for the best prompt combinations for each model and each dataset in Appendix C.

**Typology induction**: In order to analyse the data points where the models are most vulnerable we employ the following heuristics for each dataset. In these heuristics we always consider the **Vanilla + Defn + Exp (output)** setup.

For the implicit hate dataset we sort the data points in non-decreasing order based on the BERTScore between the ground truth implied statement and the generated explanation at the output. Starting from the top of this list, we consider the data points that are either not_hate in the ground truth and misclassified as implicit_hate or vice versa by **all the three** models. Note that these data points constitute the cases where the models misclassify and provide poor explanation for their classification decision. These data points are then passed through LDA (Jelodar et al., 2019) (number of topics, $K = 3$) to induce the types.

For the HateXplain and ToxicSpans datasets we sort the data points in non-decreasing order based on the sentence-BLEU scores between the ground truth rationales and the generated rationales at the output. We select 80 low-scoring data points according to the BLEU/BERTScore measure. For the HateXplain (ToxicSpans) dataset, starting from the top of this list, we consider the data points that are either normal (non_toxic) in the ground truth and misclassified as hate speech/offensive (toxic) or vice versa by all the three models. These points are then passed through LDA (number of topics, $K = 3$) to induce the types. For each topic, we choose four words which have the highest probability of association with the topic. These collected set of words are then manually coded with topic names by two researchers with long experience in hate speech research. In order to obtain a semantic clarity upon the derived topic names, a meticulous manual annotation process is undertaken. This annotation task is done by three domain experts, and notably, unanimous consensus is reached among all annotators regarding the semantic interpretation and nomenclature of each delineated topic name. We note the emerging typologies in the Table 6.

**Observations from the induced typology**: For each dataset, we note some of the interesting typologies that emerged. **(1)** For the implicit hate dataset, one curious case of misclassification (non-hate → implicit hate) is the presence of the 'racist' word, while in another category we find the model marked something as implicit hate because it has pro-white sentiments. Even when post representing news articles or opinion pieces contain sensitive words like 'blacks', 'antifa' etc., the models label them as implicit hate. **(2)** For the HateXplain dataset, we find that hateful or offensive posts get misclassified to normal due to multiple reasons – (a) presence of negations like 'neither', 'nor', 'no' etc. in the post, (b) vocabulary gap where the models do not know the usage of words like 'muzrat' used in the post, and (c) presence of polysemous words in the post with one of the meanings usually being a slang (usually not found in standard English dictionaries), e.g., 'dyke', 'furries', 'furfaggotry' etc. **(3)** For ToxicSpans dataset, we note that ideological posts and posts with fact-checking related statements about political news often get misclassified as toxic. On the other hand, posts that are implicitly toxic often get misclassified as non-toxic

| Dataset | Topic cluster | Examples | GT | PR | Type |
|---|---|---|---|---|---|
| Implicit hate | antifa, self, make, right, time, rt, memorial, requires, government, say | bring back huac rt : you can be racist all you want but socialism is another beast | n_h | imp_h | racist |
| | | these are the same people who yell about how they're not racist . this thread is so sad | n_h | imp_h | racist |
| | white, people, good, anti, nationalist, right, chief, staff, south, cruz | not enough power to make a white lol | n_h | imp_h | pro-white |
| | | how is appreciating your heritage white privilege ? we're the only ones not allowed to do it | n_h | imp_h | pro-white |
| | racist, white, america, make, people, americans, know, really, trump, new | yeah but who is defining intolerance . antifa are intolerant . | n_h | imp_h | stmt_with_sensitive_words |
| | | black #unemployment rate : 9.2 % more than double white unemployment of 4.4 % via #hireamerican | n_h | imp_h | stmt_with_sensitive_words |
| HateXplain | toxic, talk, men, masculinity, freakin, attack, nigger, slut, people, women | tbtc take back the country toxic masculinity patriarchy harm men noxious femaleness and public slut posturing empower women neither feminism nor now could have pulled this off this is the doing of taxpayer supported state university system maga | o | n | negation |
| | | note no nigger was harmed during the making of this film. | h | n | negation |
| | white, like, line, did, large, muzzies, women, tactics, scarf, counter | why he can not get a hoe that work minus the crazy | o | n | vocab_gap |
| | | massive muzrat espionage | h | n | vocab_gap |
| | common, kike, christian, better, stick, prefer, pissing, discussion, sure, provide | additionally furries can also transmit the infectious disease of furfaggotry to others | h | n | polysemy |
| | | a dyke puts the mannish moves on a femme at a lesbian bar in paris <number> photograph by george brassai | o | n | polysemy |
| ToxicSpans | trump, right, good, man, state, like, stupid, don, money, did | Right Charlie the media fabricated all those live broadcast during the primaries. Then they altered everything Trump said during the campaign. Do you have Twitter running 24/7 to keep up with Donald's latest rants? | n_t | t | ideology |
| | | The Liberals leasing the Trudeau name for their leadership is turning out to be a big, fat, failed experiment!! | n_t | t | ideology |
| | people, trump, just, know, tax, like, think, don, vote, need | The headline for this article has changed at least twice since it was originally posted yesterday. Here's the latest update: Unhinged Trump re-emerges, defending first month in White House | n_t | t | fact_check_pol_news |
| | | This article is entirely WRONG! An ongoing deficit will disintegrate the financial system AND THE COUNTRY in less than 30 years. . . . Computer projections by more than one analyst suggest a "kinetic" outcome within 15 years… | n_t | t | fact_check_pol_news |
| | just, like, make, stupid, sure, don, person, people, trump, does | Oh, gay and black, you just caused all our white christian friends here to start salivating at the same time, for what I'm not sure. | t | n_t | implicit_semantics |
| | | This individual does need therapy, I hope he gets the help he needs as well as one of those ridiculously light sex offender sentence. This system is broken when it comes to crimes of this nature. The list was designed to be a warning for parents, and yet | t | n_t | implicit_semantics |

Table 6: The typologies induced from the error cases for each dataset. GT: ground truth, PR: prediction, n_h: not_hate, imp_h: implicit_hate, o: offensive, n: normal, h: hate speech, n_t: non_toxic, t: toxic, stmt_with_sensitive_words: statement with sensitive words, vocab_gap: vocabulary gap, fact_check_pol_news: statement on fact checking information about political news.

by the models.

# 8 Conclusion

In this work, we extensively study three LLMs across three datasets on a variety of prompt setups. Overall these models, though advanced, still face lot of challenges in hate speech detection in their vanilla setting. Our prompt strategies, when applied individually, improve the performance of such models. However, merging of the prompt strategies do not help much. Last, we do a detailed error analysis and typology coding to find 'jailbreak' points where these models are vulnerable.

# 9 Limitations

We mostly focus on English datasets in this paper since we wanted to present a detailed analysis of the use of additional context (explanation, target community) which is often not present in multilingual datasets. We use these LLMs as a black box hence we do not know the inner workings of the proprietary LLMs. Lastly, although we cover a lot of well thought out prompt variations in our paper, these variations are not exhaustive.

# 10 Ethical statement

Here we discuss the ethical considerations that were not explicitly noted in the body of the paper. We use three of the LLMs `gpt-3.5-turbo-0301`, `text-davinci-003` and `flan-T5-large` to detect hate speech. These experiments were purely done from a research point of view; the actual application of such model in real world should be done with caution. This is also evident from the challenges these models face while classifying different forms of hate.

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

## A  Prompt strategies

The Table 7 shows the different prompt variants we use in this study.

## B  Definitions

### B.1  Implicit hate dataset

- **implicit_hate**: Implicit hate speech is defined by coded or indirect language that disparages a person or group on the basis of protected characteristics like race, gender, and cultural identity.
- **explicit_hate**: Explicit hate refers to openly expressed, direct forms of hatred and prejudice toward individuals or groups based on their characteristics.
- **not_hate**: This class refers to speech or actions that do not involve any form of hatred, prejudice, or discrimination toward individuals or groups based on their characteristics.

### B.2  HateXplain dataset

- **hate speech**: Any speech or text that attacks a person or group on the basis of attributes such as race, religion, ethnic origin, national origin, gender, disability, sexual orientation, or gender identity.
- **offensive**: The text or speech which uses abusive slurs or derogatory terms but may not be hate speech.
- **normal**: The text which is neither offensive or hate speech and adheres to social norms.

### B.3  ToxicSpans dataset

- **Toxic**: In social media and online forum, toxic content can be defined as rude, disrespectful, or unreasonable posts that would make users want to leave the conversation.
- **Non_toxic**: The speech or text that is not toxic and is fit for use in conversation.

| Category | Prompt templates |
|---|---|
| **Vanilla** | Classify the post delimited by three backticks as `list_of_labels`.
Provide the answer as either `list_of_labels` only.
`example_outputs`
```post``` |
| **Vanilla + Defn** | Consider the following definitions.
`list_of_defintions`
Based on this classify the post delimited by three backticks as `list_of_labels`
Provide answer in the format : either `list_of_labels` only.
`example_outputs`
```post``` |
| **Vanilla + Exp (output)** | Classify the post delimited by three backticks as `list_of_labels` and `explanation_type`.
Provide answer in the format : either `list_of_labels` only followed by `explanation_format`
`example_outputs`
```post``` |
| **Vanilla + Exp (input)** | Classify the post delimited by three backticks as `list_of_labels` taking into account the `explanation`
for why a post should be considered `list_of_labels`.
Provide answer in the format : either `list_of_labels` only.
`example_outputs`
```post``` |
| **Vanilla + Tar (output)** | Classify the post delimited by three backticks as `list_of_labels` and `target_type`
Provide answer in the format : either `list_of_labels` only followed by `target_format`
`example_outputs`
```post``` |
| **Vanilla + Tar (input)** | Classify the post delimited by three backticks as `list_of_labels` with respect to the victim community `targets` .
Provide answer in the format : either `list_of_labels` only.
`example_outputs`
```post``` |
| **Vanilla + Defn + Exp (input)** | Consider the following definitions.
`list_of_defintions`
Based on this classify the post delimited by three backticks as `list_of_labels` taking into account the `explanation`
for why a post should be considered `list_of_labels`
Provide answer in the format : either `list_of_labels` only.
`example_outputs`
```post``` |
| **Vanilla + Defn + Exp (output)** | Consider the following definitions.
`list_of_defintions`
Based on this classify the post delimited by three backticks as `list_of_labels` and `explanation_type`.
Provide answer in the format : either `list_of_labels` only followed by `explanation_format`
`example_outputs`
```post``` |
| **Vanilla + Defn + Tar (output)** | Consider the following definitions.
`list_of_defintions`
Based on this classify the post delimited by three backticks as `list_of_labels` and `target_type`
Provide answer in the format : either `list_of_labels` only followed by `target_format`
`example_outputs`
```post``` |
| **Vanilla + Defn + Tar (input)** | Consider the following definitions.
`list_of_defintions`
Based on this classify the post delimited by three backticks as `list_of_labels` with respect to the victim community `targets` .
Provide answer in the format : either `list_of_labels` only.
`example_outputs`
```post``` |

Table 7: The different prompts used in our experiments.

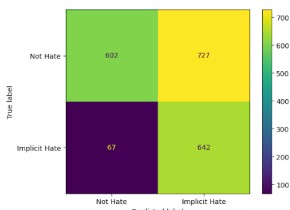

Figure 1: Vanilla + target_input for implicit hate.

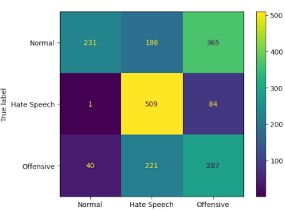

Figure 2: Vanilla + target_input for HateXplain.

## C Confusion matrix of best performing prompt combinations

### C.1 GPT-3.5

The confusion matrices for the three best prompt combinations for the three datasets in connection to the `gpt-3.5-turbo-0301` are illustrated in Figures 1, 2, and 3.

### C.2 Davinci

The confusion matrices for the three best prompt combinations for the three datasets in connection to the `text-davinci-003` are illustrated in Figures 4, 5, and 6.

### C.3 Flan-T5

The confusion matrices for the three best prompt combinations for the three datasets in connection to the `flan-T5-large` are illustrated in Figures 7, 8, and 9.

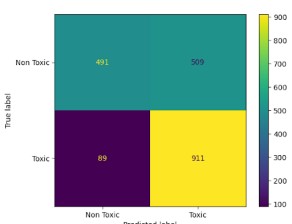

Figure 3: Vanilla + explanation_input for ToxicSpans.

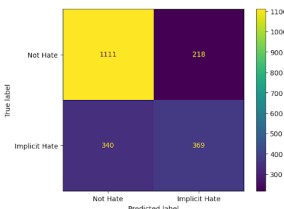

Figure 4: Vanilla + target_input for implicit hate.

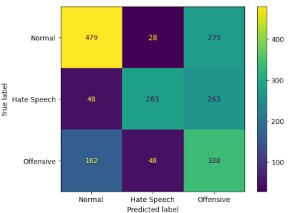

Figure 5: Vanilla + target_input for HateXplain.

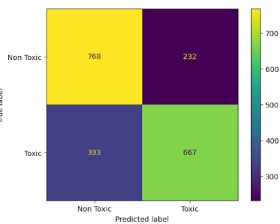

Figure 6: Vanilla + definition_input for ToxicSpans.



Figure 7: Vanilla + definition_input for implicit hate.

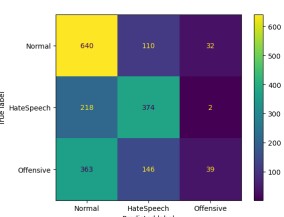

Figure 8: Vanilla + definition_input + target_input for HateXplain.

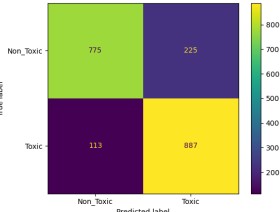

Figure 9: Vanilla + explanation_input for ToxicSpans.