# OpenReview forum: "Probing LLMs for hate speech detection: strengths and vulnerabilities"
_EMNLP/2023/Conference — EMNLP 2023 Findings_

### Official Review · Reviewer_Dc6M · 2023-08-04

**Soundness:** 3

**Excitement:**

3: Ambivalent: It has merits (e.g., it reports state-of-the-art results, the idea is nice), but there are key weaknesses (e.g., it describes incremental work), and it can significantly benefit from another round of revision. However, I won't object to accepting it if my co-reviewers champion it.

**Paper Topic And Main Contributions:**

This paper presents the authors' experience in utilizing various algorithms for hate speech detection. Through an extensive study of two LLMs (GPT 3.5 and text-davinci) across three datasets with different prompt setups, the authors reveal the challenges faced by these advanced models in detecting hate speech in their default configurations. They propose individual prompt strategies that enhance the models' performance, but find limited improvement when combining these strategies. Additionally, the authors conduct a detailed error analysis and typology coding, identifying vulnerable points ('jail-break' points) in the models' performance. This paper offers valuable insights for readers interested in the field of hate speech detection and the capabilities of language models.

**Questions For The Authors:**

- It would be valuable for the authors to explain the decision-making process behind the use of the prompting system and LLMs models. This would help researchers to understand the benefits and limitations of these models and potentially apply them to their own research.
- While the authors have provided a comprehensive overview of the hate speech detection task, additional information on the linguistic aspects of the task would be beneficial. This would help readers to understand the theoretical foundations of the study and the methods used to achieve the results.
- Data preprocessing techniques are not described despite its importance.
- Are the results reported in this paper statistically significant?

**Reasons To Accept:**

- The paper exhibits good organization and presentation overall.
- It focuses on specific benchmark datasets related to a particular task.
- The process of prompting configurations is thoroughly explained.
- The proposed approach offers a notably simpler and more efficient method for hate speech detection.
- The reported performance of the prompting system and various commonly used LLMs models on this task is valuable for future benchmarking endeavors.

**Reasons To Reject:**

- In introduction, authors said "In terms of vanilla prompts (that is case (a)), we find that text-davinci-003 is better than gpt-3.5-turbo-0301 although the latter is a more recent version". Authors are confused with problem definition and results. The introduction is for defining the problem, not demonstrating the results.
- While the authors have discussed the results of their study, they have not provided details on the data preprocessing techniques used. As data preprocessing is a crucial step in natural language processing tasks, it is important to include this information to ensure the reproducibility of the study.
- The manuscript makes no mention of hyper-parameters fine-tuning or other model configures. Please explain how these parameters are optimized.
- While the authors have provided a comprehensive overview of the hate speech detection task, additional information on the linguistic aspects of the task would be beneficial. This would help readers to understand the theoretical foundations of the study and the methods used to achieve the results.



**Reproducibility:**

3: Could reproduce the results with some difficulty. The settings of parameters are underspecified or subjectively determined; the training/evaluation data are not widely available.

**Reviewer Confidence:**

5: Positive that my evaluation is correct. I read the paper very carefully and I am very familiar with related work.

**Typos Grammar Style And Presentation Improvements:**

- Please avoid using vague terms like "many", "few" to ensure clarity and transparent in the writing.
- The authors should present the results in Tables 2, 3, 4 in a more comprehensible method. It is possible to optimise them by combining them into one.
- Authors must unify the numerical formatting (thousands separator, decimal place, alignment).
- The authors should focus on clarifying the problem, motivation, and contribution of the research in the Introduction rather than going into too much detail about the method, which produces overlap with the content in section 6 and makes the content confused.

---

> ### Author Rebuttal · Authors · 2023-08-28
>
> 1. In introduction, authors said "In terms of vanilla prompts (that is case (a)), we find that text-davinci-003 is better than gpt-3.5-turbo-0301 although the latter is a more recent version". Authors are confused with problem definition and results. The introduction is for defining the problem, not demonstrating the results
>
> Response:- Thanks. We shall clearly delineate the introduction more in the revised version. We shall have a separate paragraph outlining the contributions followed by just the highlights of the results.
>
> 2. While the authors have discussed the results of their study, they have not provided details on the data preprocessing techniques used. As data preprocessing is a crucial step in natural language processing tasks, it is important to include this information to ensure the reproducibility of the study
>
> Response:- Our datasets are directly taken from the three papers  Hatexplain: A Benchmark Dataset for Explainable Hate Speech Detection  (https://arxiv.org/abs/2012.10289), Latent Hatred: A Benchmark for Understanding Implicit Hate Speech (https://arxiv.org/abs/2109.05322) and ToxiSpanSE: An Explainable Toxicity Detection in Code Review Comments (https://arxiv.org/abs/2307.03386) where the authors have described the pre-processing in extensive detail. Only for the Implicit Hate dataset, we selected the test set in a stratified fashion. Everything else is verbatim, the same as the original source papers. We shall clarify this in the revised version.
>
> 3. The manuscript makes no mention of hyper-parameters fine-tuning or other model configures. Please explain how these parameters are optimized.
>
> Response:- Please note that the experiments were run on OpenAI black box models where it is not possible to fine-tune the hyperparameters. The temperature coefficient was kept at 0 all through.
>
> 4. While the authors have provided a comprehensive overview of the hate speech detection task, additional information on the linguistic aspects of the task would be beneficial. This would help readers to understand the theoretical foundations of the study and the methods used to achieve the results.
>
> Response:- Thanks. Indeed, this is exactly why we conducted the typology induction, which precisely underpins the linguistic reasons for the failure of the LLMs. The entire typology induction idea was to have linguistic backing to the theoretical foundations of the LLMs. We shall clarify this in the revised version.
>
> Questions
> 1. It would be valuable for the authors to explain the decision-making process behind the use of the prompting system and LLMs models. This would help researchers to understand the benefits and limitations of these models and potentially apply them to their own research.
>
> Response:- Explaining the precise decision-making mechanism of the employed LLMs is unattainable due to their inherently opaque nature (black box). The selection of prompts, however, is underpinned by meticulous scrutiny of the datasets and the prevalent concerns within the hate speech research domain. The typology section introduces an innovative approach to comprehending LLMs' vulnerabilities, even without the insights into the internal functioning of these LLMs.
>
> 2. While the authors have provided a comprehensive overview of the hate speech detection task, additional information on the linguistic aspects of the task would be beneficial. This would help readers to understand the theoretical foundations of the study and the methods used to achieve the results
>
> Response:- Thanks. Indeed, this is exactly why we conducted the typology induction, which precisely underpins the linguistic reasons for the failure of the LLMs. The entire typology induction idea was to have linguistic backing to the theoretical foundations of the LLMs. We shall clarify this in the revised version.
>
> 3. Data preprocessing techniques are not described despite their importance.
>
> Response:- Our datasets are directly taken from the three papers  Hatexplain: A Benchmark Dataset for Explainable Hate Speech Detection  (https://arxiv.org/abs/2012.10289), Latent Hatred: A Benchmark for Understanding Implicit Hate Speech (https://arxiv.org/abs/2109.05322) and ToxiSpanSE: An Explainable Toxicity Detection in Code Review Comments (https://arxiv.org/abs/2307.03386)where the authors have described the pre-processing in details. Only for the Implicit Hate dataset, we selected the test set in a stratified fashion. Everything else is verbatim, the same as the original source papers. We shall clarify this in the revised version.
>
> 4. Are the results reported in this paper statistically significant?
>
> Response:- Please note that performance enhancement is not an objective of this paper. Rather our main objective here was to AUDIT (https://hci.stanford.edu/publications/2021/FnT_AuditingAlgorithms.pdf) SOTA LLMs against various prompt injections. Nevertheless, as per your suggestion, we compared the results of the vanilla model for each dataset and the best prompt with the model using the Mann-Whitney U test. We find that the results are significant. We present the p-values for each.
>
> Chatgpt-HX : p-value=0.0042 (vanilla vs target-at-in)
>
> Chatgpt-IH : p-value=0 (vanilla vs target-at-in)
>
> Chatgpt-TS : p-value=0.00088 (vanilla vs explain-at-in)
>
> DaVinci-HX : p-value=9.946e-11 (vanilla vs target-at-in)
>
> DaVinci-IH : p-value=4.441e-16 (vanilla vs target-at-in)
>
> DaVinci-TS : p-value=0.01524 (vanilla vs def-at-in)

---

### Official Review · Reviewer_UTwD · 2023-08-06

**Soundness:** 3

**Excitement:**

3: Ambivalent: It has merits (e.g., it reports state-of-the-art results, the idea is nice), but there are key weaknesses (e.g., it describes incremental work), and it can significantly benefit from another round of revision. However, I won't object to accepting it if my co-reviewers champion it.

**Paper Topic And Main Contributions:**

The paper performs various probing experiments to analyze how well the latest LLM systems work regarding hate speech detection.  The authors then introduce a typology for the kind of texts that is challenging to classify and can be employed for adversarial purposes.

**Questions For The Authors:**

1. For analysing typology of attack, why was the explanation model picked over other?
2. If the prompting is done in a CoT manner rather than a single prompt, do the results still remain the same?
3. For the case where inclusion of definitions increases the error, could it be cause the model then starts looking for hate where non exists?


**Reasons To Accept:**

1. The study encompass 3 datasets and 6 prompt designs which appears to be a good starting point for analysing SOTA LLMs for hate speech detection.
2. The authors highlight the kind of adversarial attacks where the LLM systems fail to detect hate.

**Reasons To Reject:**

1. While the introduction of typology for adversarial/jailbreak techniques to bypass hate speech is a good starting point, the authors have not shared the details on how the typology was developed. The definitions and scope of these definitions, annotation guidelines, and agreement scores obtained during annotation must be included. Adapting and extending the typology for future use cases will be challenging without these details.
2. The paper needs significant reformatting--> expand on the error analysis and discussion of the adversarial attacks and the paper title and abstract emphasis on the "vulnerability aspect" as well. Similarly, there needs to be more details about the hyper parameters used (default or otherwise), making reproducibility challenging.
3. The exact motivation and application of the prompts where the target and explanation information are given in the input is unclear. In the real world, such information is not available initially; therefore, the use case and the promising results thus obtained from this prompt design are not helpful in the real world. This is a significant concern for me.
4. The authors should also include significant tests to corroborate the results in Tables 2,3,4. Further, the justification for employing a smaller subset from the three datasets needs to be provided. For example, there are multiple types of implicit hate in the implicit hate dataset. Were the samples stratified based on these labels or only on the first level of labels (implicit, explicit, and non-hate)? It will be better if multiple subsets or the whole dataset is considered for analysis.
5. Further, existing benchmark results on transformer based hate speech detection system or whatever is the SOTA for the respective datasets. This will help in understanding the delta improvement in performance that updated GPT systems. This will also help error analysis determine which kind of hate speech is now easier to classify (i,e misclassified by existing baselines but rectified by SOTA GPT systems), vs the ones that still remain challenging. These challenging samples can be employed to form a more concrete set of jailbreak samples.
6. Before starting with sending the definitions along with text, it would be better if we first establish the definitions that such systems can generate. This will help establish what these LLM models in general consider hateful.
7. An error analysis on the generated explanation/target group is missing and vaguely discussed. How much of these results are coherent vs hallucinated. Are they any target groups that are less accurate detected. Are the explanations coherent and can they be utilised for content moderation?

**Reproducibility:**

2: Would be hard pressed to reproduce the results. The contribution depends on data that are simply not available outside the author's institution or consortium; not enough details are provided.

**Reviewer Confidence:**

4: Quite sure. I tried to check the important points carefully. It's unlikely, though conceivable, that I missed something that should affect my ratings.

---

> ### Author Rebuttal · Authors · 2023-08-28
>
> 1. While the introduction of typology for adversarial/jailbreak techniques to bypass hate speech is a good starting point, the authors have not shared the details on how the typology was developed. The definitions and scope of these definitions, annotation guidelines, and agreement scores obtained during annotation must be included. Adapting and extending the typology for future use cases will be challenging without these details.
>
> Response:- To develop the typology of adversarial/jailbreak techniques, we first selected the cases where the models produced erroneous outputs and then selected 80 low-scoring data points according to the bleu/bleurt score measure over the explanation generated by these models. Then, we applied Latent Dirichlet Allocation (LDA) to this subset of low-scoring data points. The number of topics in each case is determined by setting K=3, a value chosen based on the coherence score. Next we manually examine the top 10 most probable words encapsulated within each topic cluster. In order to obtain a semantic clarity upon the derived topics, a meticulous manual annotation process was undertaken. This annotation task was conducted by three domain experts, and notably, unanimous consensus was reached among all annotators regarding the semantic interpretation and nomenclature of each delineated topic cluster. This noteworthy unanimity among annotators is pivotal and will be explicitly elucidated in the forthcoming revised iteration of this manuscript.
>
> 2. The paper needs significant reformatting--> expand on the error analysis and discussion of the adversarial attacks and the paper title and abstract emphasis on the "vulnerability aspect". Similarly, there needs to be more details about the hyperparameters used (default or otherwise), making reproducibility challenging
>
> Response:- We have detailed confusion matrices for each dataset (see below), each prompt type, and each model used. We shall report all of these in the revised version.
>
> 3. The exact motivation and application of the prompts where the target and explanation information are given in the input is unclear. In the real world, such information is not available initially; therefore, the use case and the promising results thus obtained from this prompt design are not helpful in the real world. This is a significant concern for me.
>
> Response:- The primary objective of this study was to conduct an exhaustive analysis of the efficacy of distinct prompt injections and their inherent susceptibilities. Given that a multitude of datasets encompass details pertaining to explanations and targets, it was imperative not to disregard a subset of prompt injections wherein these specific details were overtly incorporated within the prompts. Consequently, any enhancement in performance observed in the context of these prompt injections is regarded as an ancillary outcome, albeit one that we deemed essential to highlight.
>
> 4. The authors should also include significant tests to corroborate the results in Tables 2,3,4. Further, the justification for employing a smaller subset from the three datasets must be provided. For example, there are multiple types of implicit hate in the implicit hate dataset. Were the samples stratified based on these labels or only on the first level of labels (implicit, explicit, and non-hate)? It will be better if multiple subsets or the whole dataset is considered for analysis.
>
> Response:- Please note that performance enhancement is not an objective of this paper. Rather our main objective hee was to AUDIT (https://hci.stanford.edu/publications/2021/FnT_AuditingAlgorithms.pdf) SOTA LLMs against various prompt injections. Nevertheless, as per your suggestion, we compared the results of the vanilla model for each dataset and the best prompt with the model using the Mann-Whitney U test. We find that the results are significant. We present the p-values for each.
>
> Chatgpt-HX : p-value=0.0042 (vanilla vs target-at-in)
>
> Chatgpt-IH : p-value=0 (vanilla vs target-at-in)
>
> Chatgpt-TS : p-value=0.00088 (vanilla vs explain-at-in)
>
> DaVinci-HX : p-value=9.946e-11 (vanilla vs target-at-in)
>
> DaVinci-IH : p-value=4.441e-16 (vanilla vs target-at-in)
>
> DaVinci-TS : p-value=0.01524 (vanilla vs def-at-in)
>
> 5. The subset of the dataset was selected using the samples stratified based on the first level of labels. Going further below made the exploration space sparse.
>
> Further, existing benchmark results on transformer based hate speech detection system or whatever is the SOTA for the respective datasets. This will help in understanding the delta improvement in performance that updated GPT systems. This will also help error analysis determine which kind of hate speech is now easier to classify (i,e misclassified by existing baselines but rectified by SOTA GPT systems), vs the ones that still remain challenging. These challenging samples can be employed to form a more concrete set of jailbreak samples.
>
> Response:- BERT-HateXplain, a state-of-the-art model has performed better than gpt-3.5 (34.23%) and text-davinci (22.46%) on HateXplain dataset, but it hasn’t performed as well when we inference upon other datasets. Once again, our objective was not to explicitly demonstrate delta improvements of LLMs but to perform an audit study to understand the capabilities of these models.
>
> 6. Before starting with sending the definitions along with text, it would be better if we first establish the definitions that such systems can generate. This will help establish what these LLM models in general consider hateful.
>
> Response:- The definitions provided by the LLMs on hate speech or offensive speech, or implicit hate speeches cannot be used to investigate the efficacy of the LLMs.The definitions provided by the authors of the papers containing the datasets are used for the annotations of the data. Hence it will be erroneous to use any other definition other than the verbatim definitions as given in the papers referenced for the datasets.
>
> 7. An error analysis on the generated explanation/target group is missing and vaguely discussed. How much of these results are coherent vs hallucinated. Are there any target groups that are less accurate detected. Are the explanations coherent and can they be utilised for content moderation?
>
> 	Response:- Please find below the confusion matrices for the vanilla vs best prompts for each model and each dataset combination. We shall add these results in the revised version of the manuscript.
>
> ChatGPT-IH-Vanilla:
> |               | Not-Hate | Explicit-Hate | Implicit-Hate |
> | ------------- | -------- | ------------- | ------------- |
> | Not-Hate      | 358      | 273           | 698           |
> | Explicit-Hate | 0        | 94            | 14            |
> | Implicit-Hate | 27       | 394           | 289           |
>
> ChatGPT-IH-Target-in: (improvement in classification of the Not_hate class)
> |               | Not-Hate | Implicit-Hate |
> | ------------- | -------- | ------------- |
> | Not-Hate      | 602      | 727           |
> | Implicit-Hate | 67       | 642           |
>
> ChatGPT-HX-Vanilla:
> |            | Normal | Hatespeech | Offensive |
> | ---------- | ------ | ---------- | --------- |
> | Normal     | 192    | 388        | 202       |
> | Hatespeech | 2      | 575        | 17        |
> | Offensive  | 19     | 433        | 96        |
>
> ChatGPT-HX-Target-in: (improvement in classification of the Offensive class)
> |            | Normal | Hatespeech | Offensive |
> | ---------- | ------ | ---------- | --------- |
> | Normal     | 231    | 186        | 365       |
> | Hatespeech | 1      | 509        | 84        |
> | Offensive  | 40     | 221        | 287       |
>
> ChatGPT-TS-vanilla:
> |           | Non-Toxic | Toxic |
> | --------- | --------- | ----- |
> | Non-Toxic | 473       | 527   |
> | Toxic     | 92        | 908   |
>
> ChatGPT-TS-Explan-in: (improvement on both classes)
> |           | Non-Toxic | Toxic |
> | --------- | --------- | ----- |
> | Non-Toxic | 491       | 509   |
> | Toxic     | 89        | 911   |
>
> Davinci-IH-vanilla:
> |               | Not-Hate | Explicit-Hate | Implicit-Hate |
> | ------------- | -------- | ------------- | ------------- |
> | Not-Hate      | 994      | 50            | 285           |
> | Explicit-Hate | 317      | 46            | 347           |
> | Implicit-Hate | 15       | 7             | 86            |
>
> Davinci-IH-target-in: (improvement on both classes)
> |               | Not-Hate | Implicit-Hate |
> | ------------- | -------- | ------------- |
> | Not-Hate      | 1111     | 218           |
> | Implicit-Hate | 340      | 369           |
>
> Davinci-HX-vanilla:
> |            | Normal | Hatespeech | Offensive |
> | ---------- | ------ | ---------- | --------- |
> | Normal     | 418    | 35         | 39        |
> | Hatespeech | 47     | 114        | 433       |
> | Offensive  | 145    | 30         | 373       |
>
> Davinci-HX-target-in: (improvement on the Hatespeech class)
> |            | Normal | Hatespeech | Offensive |
> | ---------- | ------ | ---------- | --------- |
> | Normal     | 418    | 28         | 275       |
> | Hatespeech | 48     | 283        | 263       |
> | Offensive  | 162    | 48         | 338       |
>
> Davinci-TS-vanilla:
> |           | Non-Toxic | Toxic |
> | --------- | --------- | ----- |
> | Non-Toxic | 796       | 204   |
> | Toxic     | 381       | 619   |
>
> Davinci-TS-Def-in:( improvement on the Toxic class)
> |           | Non-Toxic | Toxic |
> | --------- | --------- | ----- |
> | Non-Toxic | 768       | 232   |
> | Toxic     | 333       | 667   |
>
>
> Questions
>
> 1. For analysing the typology of attack, why was the explanation model picked over other?
>
> We require cases in which the model makes incorrect classifications and provides poor explanations for its classification decisions. This can be achieved by instructing the model to generate explanations, which can then be compared against the ground truth explanations. This comparative analysis allows for a quantitative assessment of the quality of the generated explanations; hence, the explanation model was chosen.
>
> 2. If the prompting is done in a CoT manner rather than a single prompt, do the results still remain the same?
>
> While we worked for the fully zero-shot setting, your suggestion of presenting the prompts in a CoT fashion is interesting and we look forward to taking this up as a future work.
>
> 3. For the case where inclusion of definitions increases the error, could it be cause the model then starts looking for hate where non exists
>
> Since it is a black box model, it is difficult to comment on the exact workings of the model. Intuitively, we also feel that something similar should be taking place.

---

### Official Review · Reviewer_xYYx · 2023-08-07

**Paper Topic And Main Contributions:** 1. The paper is about probing large l…
**Typos Grammar Style And Presentation Improvements:** 1. The writing style of the paper is …
**Soundness:** 4

**Excitement:**

3: Ambivalent: It has merits (e.g., it reports state-of-the-art results, the idea is nice), but there are key weaknesses (e.g., it describes incremental work), and it can significantly benefit from another round of revision. However, I won't object to accepting it if my co-reviewers champion it.

**Missing References:**

N/A

**Reasons To Accept:**

1. The paper addresses an important and timely issue of detecting hate speech using large language models, which is relevant to both academia and industry.
2. The paper presents a comprehensive analysis of the performance of two LLMs (GPT-3.5 and text-davinci) in detecting hate speech, and explores the impact of additional context on the performance of LLMs, outlining multiple prompt stragies.
3. The paper identifies the vulnerabilities and limitations of the two LLMs in detecting hate speech, and conduct a detailed error analysis and typology coding to find ‘jail-break’ points where these models are vulnerable.
4. The paper provides valuable insights and recommendations for future research in this area.

**Reasons To Reject:**

1. While the paper addresses an important issue, the scope of the study is limited to only two large language models and three datasets. This may not be sufficient to draw generalizable conclusions about the performance of LLMs in detecting hate speech.
2. The paper does not provide a detailed explanation of the methodology used to select the datasets and the evaluation metrics. This may raise questions about the validity and reliability of the results.
3. The paper does not provide a clear definition of hate speech, which may lead to ambiguity in the interpretation of the results.
4. The paper does not consider other valid baselines and SoTA models while reporting the results. Some baselines to include could be: BERT-HateXplain (LIME and Attention) by Binny et al., AAAI 2021 and BERT-RP/ BERT-MRP by Kim et al., COLING 2022.
6. The presentation of the paper is generally well-organized, but there are some areas where the structure could be improved. For example, the introduction could provide more context and motivation for the study, and the methodology section could provide more detail on the selection of datasets and evaluation metrics.

**Reproducibility:**

4: Could mostly reproduce the results, but there may be some variation because of sample variance or minor variations in their interpretation of the protocol or method.

**Reviewer Confidence:**

4: Quite sure. I tried to check the important points carefully. It's unlikely, though conceivable, that I missed something that should affect my ratings.

---

> ### Author Rebuttal · Authors · 2023-08-28
>
> 1. While the paper addresses an important issue, the scope of the study is limited to only two large language models and three datasets. This may not be sufficient to draw generalizable conclusions about the performance of LLMs in detecting hate speech.
>
> Response:- This paper introduces a framework for the zero-shot evaluation of hate speech detection with LLMs. We chose two of the popular LLMs - gpt-3.5-turbo and text-davinci for our evaluation. The datasets are benchmark datasets in hate speech detection and further have explanations/rationales that can be used to understand the models' behavior. As per your suggestion, we have further extended our study to FlanT5-large and found comparable results. We have added the performance table below. We shall also add these new results in the revised manuscript.
>
> Model:FLAN-T5-Large(783M):
>
> Toxic Span:
>
> |           | nd_ne | nd_eo  | nd_ei | d_ne  | d_eo  | d_ei   |
> | --------- | ----- | ------ | ----- | ----- | ----- | ------ |
> | Accuracy  | 0.704 | 0.781  | 0.831 | 0.797 | 0.823 | 0.8255 |
> | Precision | 0.762 | 0.842  | 0.835 | 0.810 | 0.842 | 0.828  |
> | Recall    | 0.704 | 0.7815 | 0.831 | 0.796 | 0.823 | 0.825  |
> | F1 Score  | 0.686 | 0.823  | 0.830 | 0.794 | 0.823 | 0.825  |
>
> Implicit Hate:
>
> |           | nd_ne | nd_eo   | nd_ei | nd_to | d_ne  | d_eo  | d_ei  | d_to  | d_ti  |
> | --------- | ----- | ------- | ----- | ----- | ----- | ----- | ----- | ----- | ----- |
> | Accuracy  | 0.667 | 0.661   | 0.639 | 0.632 | 0.688 | 0.670 | 0.637 | 0.666 | 0.624 |
> | Precision | 0.643 | 0.655   | 0.608 | 0.639 | 0.667 | 0.656 | 0.617 | 0.642 | 0.558 |
> | Recall    | 0.624 |   0.582 | 0.561 | 0.647 | 0.658 | 0.663 | 0.544 | 0.612 | 0.545 |
> | F1 Score  | 0.626 | 0.564   | 0.542 | 0.628 | 0.661 | 0.658 | 0.502 | 0.613 | 0.541 |
>
> Hatexplain:
>
>
> |           | nd_ne | nd_eo | nd_ei | d_eo  | d_ei  | d_ne  | nd_to | d_to  | d_ti  |
> | --------- | ----- | ----- | ----- | ----- | ----- | ----- | ----- | ----- | ----- |
> | Accuracy  | 0.600 | 0.542 | 0.604 | 0.569 | 0.535 | 0.506 | 0.641 | 0.618 | 0.624 |
> | Precision | 0.701 | 0.634 | 0.624 | 0.568 | 0.688 | 0.666 | 0.702 | 0.605 | 0.661 |
> | Recall    | 0.653 | 0.596 | 0.624 | 0.570 | 0.603 | 0.578 | 0.566 | 0.606 | 0.654 |
> | F1 Score  | 0.589 | 0.526 | 0.604 | 0.565 | 0.505 | 0.465 | 0.517 | 0.606 | 0.623 |
>
>
> 2. The paper does not provide a detailed explanation of the methodology used to select the datasets and the evaluation metrics. This may raise questions about the validity and reliability of the results.
>
> Response:- As mentioned in the previous answer, the choice of the datasets is driven by the presence of rationales/explanations. The datasets that we have chosen are the only benchmark ones known to us that have rationales/explanations along with class labels. Further, these datasets are popular and well-cited in the literature. We selected the evaluation metrics from Table 6 in HateXplain: A Benchmark Dataset for Explainable Hate Speech Detection  (https://arxiv.org/abs/2012.10289) and  Latent Hatred: A Benchmark for Understanding Implicit Hate Speech (https://arxiv.org/abs/2109.05322) papers.
>
> 3. The paper does not provide a clear definition of hate speech, which may lead to ambiguity in the interpretation of the results.
>
> Response:- The definitions of implicit hate speech, hate speech, offensive speech, and toxic speech are verbatim adapted from the papers introducing these datasets as mentioned in Section 3: Datasets and metrics. Thus there is no one single definition and it (slightly) varies as per the authors introducing the three datasets. We shall clarify this in the revised version.
>
> 4. The paper does not consider other valid baselines and SoTA models while reporting the results. Some baselines to include could be: BERT-HateXplain (LIME and Attention) by Binny et al., AAAI 2021 and BERT-RP/ BERT-MRP by Kim et al., COLING 2022.
>
> Response:- As per your suggestion, as an additional baseline, we considered the BERT-HateXplain and ran the model on the Toxic spans and Implicit hate dataset. We already have the results for the HateXplain dataset taken from the original paper. These results are presented below and will be added in the revised manuscript.
>
> | Dataset              | F1-Score | Precision | Recall |
> | -------------------- | -------- | --------- | ------ |
> | Hatexplain           | 0.698    | 0.697     | NA     |
> | Implicit Hate (full) | 0.54     | 0.64      | 0.47   |
> | Toxic Span           | 0.31     | 0.79      | 0.20   |

---

### Meta-Review · Area_Chair_4xSM · 2023-09-15

**Recommendation:** 3

**Metareview:**

This paper delves into the efficacy of large language models (LLMs) in hate speech detection. Through a comprehensive analysis of two prominent LLMs (GPT 3.5 and text-davinci) across distinct datasets, the authors evaluate their performance and the impact of various context cues, such as explanations and targeted communities. The research unveils the models' inherent challenges in default configurations and introduces tailored prompt strategies for performance optimization.

The advantages of the paper are: 1) addresses an important and timely issue of detecting hate speech using large language models. 2) Presented comprehensive analysis over two commercially available LLMs 3) Identifies the vulnerabilities and limitations such as vulnerability to adversarial attacks. 4) Provide a good benchmark for follow up research

Some concerns were also raised regarding 1) The definition of hate speech is not clear and result in ambiguity in the interpretation of the results 2) the technical detail of the approach is not clearly described (e.g. data preprocessing techniques, hyper-parameters ) 3) The experiment is only limited to two LLMs that form a same family (GPT3) ); In addition 4) the paper needs significant reformatting

---

### Decision · Program_Chairs · 2023-10-07

**Decision:**

Accept-Findings

**Comment:**

This paper delves into the efficacy of large language models (LLMs) in hate speech detection. Through a comprehensive analysis of two prominent LLMs (GPT 3.5 and text-davinci) across distinct datasets, the authors evaluate their performance and the impact of various context cues, such as explanations and targeted communities. The research unveils the models' inherent challenges in default configurations and introduces tailored prompt strategies for performance optimization.

The advantages of the paper are: 1) addresses an important and timely issue of detecting hate speech using large language models. 2) Presented comprehensive analysis over two commercially available LLMs 3) Identifies the vulnerabilities and limitations such as vulnerability to adversarial attacks. 4) Provide a good benchmark for follow up research

Some concerns were also raised regarding 1) The definition of hate speech is not clear and result in ambiguity in the interpretation of the results 2) the technical detail of the approach is not clearly described (e.g. data preprocessing techniques, hyper-parameters ) 3) The experiment is only limited to two LLMs that form a same family (GPT3) ); In addition 4) the paper needs significant reformatting